# Effects of Complete and Partial Loss of the 24S-Hydroxycholesterol-Generating Enzyme *Cyp46a1* on Behavior and Hippocampal Transcription in Mouse

**DOI:** 10.3390/biom14030254

**Published:** 2024-02-21

**Authors:** Hong-Jin Shu, Luke H. Ziolkowski, Sofia V. Salvatore, Ann M. Benz, David F. Wozniak, Carla M. Yuede, Steven M. Paul, Charles F. Zorumski, Steven Mennerick

**Affiliations:** 1Department of Psychiatry, Washington University in St. Louis School of Medicine, St. Louis, MO 63110, USAsofias@wustl.edu (S.V.S.); wozniakd@wustl.edu (D.F.W.); yuedec@wustl.edu (C.M.Y.); smpaul@wustl.edu (S.M.P.);; 2Taylor Family Institute for Innovative Psychiatry Research, Washington University in St. Louis School of Medicine, St. Louis, MO 63110, USA

**Keywords:** 24S-hydroxycholesterol, oxysterol, liver X receptor, hippocampus, learning

## Abstract

Brain cholesterol metabolic products include neurosteroids and oxysterols, which play important roles in cellular physiology. In neurons, the cholesterol oxidation product, 24S-hydroxycholesterol (24S-HC), is a regulator of signaling and transcription. Here, we examined the behavioral effects of 24S-HC loss, using global and cell-selective genetic deletion of the synthetic enzyme CYP46A1. Mice that are globally deficient in CYP46A1 exhibited hypoactivity at young ages and unexpected increases in conditioned fear memory. Despite strong reductions in hippocampal 24S-HC in mice with selective loss of CYP46A1 in VGLUT1-positive cells, behavioral effects were not recapitulated in these conditional knockout mice. Global knockout produced strong, developmentally dependent transcriptional effects on select cholesterol metabolism genes. These included paradoxical changes in Liver X Receptor targets. Again, conditional knockout was insufficient to recapitulate most changes. Overall, our results highlight the complex effects of 24S-HC in an in vivo setting that are not fully predicted by known mechanisms. The results also demonstrate that the complete inhibition of enzymatic activity may be needed for a detectable, therapeutically relevant impact on gene expression and behavior.

## 1. Introduction

There is interest in exploiting cholesterol metabolism pathways, including neurosteroids and oxysterols, for the treatment of brain disorders, such as neurodegenerative illnesses [1,2,3,4] and neuropsychiatric disorders [5,6]. An important target of these efforts is the main cholesterol metabolizing enzyme in the brain, CYP46A1, which produces the oxidation product 24S-hydroxycholesterol (24S-HC) [7,8]. 24S-HC is the major path by which the brain eliminates excess cholesterol and was subsequently discovered to be a positive allosteric modulator of NMDA receptor function and a modulator of neuronal sensitivity to energy deprivation [5,9,10,11,12]. Early work showed profound hippocampal learning and plasticity deficits in middle-aged mice constitutively lacking *Cyp46a1* [13]. In contrast, our previous work found that LTP was largely intact with the constitutive loss of *Cyp46a1*, but some forms of NMDA receptor-dependent LTD were deficient [9,14]. Taken together, these observations have led to interest in targeting CYP46A1 for cognitive therapy and therapy for neurodegenerative illness. 

The effects of CYP46A1/24S-HC are likely to involve transcriptional alterations in the brain, which, alongside the neuromodulatory affects above, could have implications for behavior. Most notably, 24S-HC is thought to induce transcription by binding the Liver X Receptor (LXR) [15,16]. Previous work has shown that the loss of *Cyp46a1* has profound transcriptional effects [17], particularly on genes involved in cholesterol homeostasis, but a lack of compensatory changes to cholesterol or to other cholesterol metabolites [18,19]. Transcriptional changes include the paradoxical regulation of genes believed to be direct targets of LXR [17,20]. Remaining questions include the impact of the partial loss of 24S-HC. CYP46A1 is believed to be mainly a neuronal enzyme [21], but the impact of cell-type selective deletion is unknown because conditional deletion models have not previously been available. 

Our work revisits prior studies by comparing the impact of the global constitutive deletion of the *Cyp46a1* gene (KO mouse), as previously utilized, and a new conditional (floxed) mouse line in which *Cyp46a1* is deleted in VGLUT1-positive excitatory neurons throughout the brain (cKO mouse). Despite the loss of most hippocampal 24S-HC with *Cyp46a1* deletion in excitatory neurons, behavioral alterations and transcriptional changes observed in the global knockout were mostly not recapitulated. Overall, our results differ from some previous findings but bolster the general idea that CYP46A1 may be a therapeutic target that modulates cognition, particularly in emotional contexts. 

## 2. Materials and Methods

### 2.1. Mice

Procedures for animal care and experiments were consistent with National Institute of Health guidelines and were approved by the Washington University Institutional Animal Care and Use Committee. Studies used the ARRIVE guidelines for reporting experiments involving animals. Unless otherwise noted, mouse age was near postnatal day (PND) 100 (range 60–140 days postnatal). Younger mice (PND 45) and older mice (PND 300–500) were used for some experiments where indicated. For ease of reference, mice are referred to as PND45, PND100, and PND400. For all behavioral, biochemistry, and transcriptional experiments, cohorts comprising mice of all genotypes/experimental groups were processed simultaneously to reduce the impact of extraneous cohort-related variables on outcome measures. 

*Cyp46a1*^−/−^ mice were obtained from Jackson Labs on mixed background (RRID:IMSR_JAX:017759) and bred for at least 5 generations onto a C57Bl/6 background for current studies using speed congenics. Conditional deletions were engineered using the Easi-CRISPR method [22,23]. Briefly, CRISPR gRNAs for in vitro testing were identified using CRISPOR (http://crispor.tefor.net/, accessed on 24 January 2018) and synthesized as gBlocks (IDT). In vitro target-specific gRNA cleavage activity was validated by transfecting N2A cells with PCR-amplified gRNA gblock and Cas9 plasmid DNA (px330, Addgene, Watertown, MA) using ROCHE Xtremegene HP. Cell pools were harvested 48 h later for genomic DNA prep, followed by Sanger sequencing of PCR products spanning the gRNA/Cas9 cleavage site and TIDE analysis (https://tide.nki.nl/, accessed on 28 March 2018) of sequence trace files. CRISPR sgRNAs and Cas9 protein for injection were purchased from IDT and complexed to generate the ribonucleoprotein (RNP) for injection. A long, single- stranded donor DNA comprising 100nt homology arms and ~800 nucleotides of *Cyp46a1* sequence spanning exon 3 and partial flanking intron sequences was synthesized (IDT). Injection concentrations were 50 ng/µL Cas9, 20 ng/µL each gRNA, and 10 ng/µL donor DNA. One-cell fertilized embryos were injected into the pronucleus and cytoplasm of each zygote. Microinjections and mouse transgenesis experiments were performed as described previously. Founder genotyping was performed by deep sequencing (MiSeq, Illumina, San Diego, CA, USA). Mosaic founders were crossed to WT to generate heterozygous F1 offspring. F1 offspring were deep-sequenced to confirm correctly targeted alleles. Floxed mice were bred to VGLUT1-cre driver line (Jackson Labs’ Stock No: 023527).

### 2.2. Behavioral Testing

For locomotor activity, mice were evaluated over a 1 h period in a transparent enclosure (47.6 × 25.4 × 20.6 cm) surrounded by a lower frame containing 4 × 8 matrices of photobeam pairs and an upper frame to quantify activity in X, Y, and Z planes. Horizontal and vertical beam breaks were automatically classified as ambulations and rearings, respectively (MotorMonitor, Hamilton-Kinder, LLC, Poway, CA, USA) in a 33 × 11 cm central zone and a 5.5 cm peripheral zone similar to previously described methods [24]. For total ambulations in the locomotor activity assay, an ambulation was recorded using an algorithm that referenced a new beam block occurring after an anchor beam in a particular X-Y dimension was released or cleared. The anchor beam refers to one of the original beams blocked in that dimension. Thus, the animal must relocate its whole body to count as an ambulation. X and Y ambulations were combined in the calculation of total ambulations. SGE-301 (20 mg/kg or vehicle, ip), a synthetic analogue of 24S-HC with good in vivo pharmacokinetic properties [11], was administered 60 min prior to locomotor testing. Animals were moved directly from locomotor testing to conditioned fear acquisition for assessing drug actions on memory acquisition. SGE-301 testing was limited to locomotion and conditioned fear at one age because of compound availability. 

Fear conditioning was evaluated as previously described [24,25]. Experimenters blinded to genotype habituated and tested mice in an acrylic chamber (26 × 18 high × 18 cm) with a metal grid floor, an LED bulb, and an inaccessible peppermint odorant. The LED was lit at trial initiation and remained illuminated. Testing on day 1 was 5 min long during which an 80 dB white noise tone sounded for 20 s at 100 s, 160 s and 220 s. A 1.0 mA shock (unconditioned stimulus; UCS) was paired with the last 2 s of the tone (conditioned stimulus; CS). Baseline freezing behavior during the first 2 min and the freezing behavior (conditioned response; CR) during the last 3 min was quantified through the image analysis software FreezeFrame (Actimetrics, Evanston, IL, USA). The testing session on day 2 lasted 8 min. The light was illuminated during the entire trial with no tones or shocks presented. This permitted the evaluation of CR to the contextual cues associated with the shock UCS from day 1. Testing on day 3 lasted 10 min, and the context of the chamber was changed to an opaque acrylic-walled chamber with a different (coconut) odorant. The CS began at 120 s and persisted to the end of the trial. Baseline freezing to the new context (pre-CS) was quantified during the first 2 min. The CR to the auditory CS associated with the shock UCS was quantified during the remaining 8 min. Shock sensitivity was evaluated following testing as previously described [24].

Morris Water Maze testing was performed on mice in a 120 cm-diameter pool of four equal-sized quadrants. Mice were exposed to two days of cued trials before beginning spatial learning trials two days later. During cued trials, the platform location was identified by a tennis ball atop a steel rod affixed to the platform. Mice were started in the quadrant opposite the goal quadrant in the cued trials. Mice were tested with four trials per day, and the platform quadrant location was varied for each trial. Place trials followed the cued trials two days later for five days during which the escape quadrant containing a submerged platform, and extra-maze cues (black shapes on the walls), which aided navigation and learning, were present. Mice were started in a different quadrant using a random start pattern. Mice were allowed to remain on the escape platform for 30 s. In cued and place trials, if the mouse did not find the platform within 60 s, it was led to the platform. Two 60 s probe trials were conducted 1 h and 48 h following the final place trial. For probe trials, the platform was removed, and time spent in the goal and other pool quadrants were analyzed. All Morris Water Maze trials were evaluated using Any-Maze video tracking system (Stoelting Co., Wood Dale, IL, USA). Behavioral data were analyzed using Prism 9.3.0 (GraphPad, La Jolla, CA) software by *t*-tests, Factorial, or Repeated Measures ANOVA. With significant main effects or interactions, post hoc comparisons were evaluated using appropriate corrections for multiple comparisons. All behavioral tests were performed by an experimenter blinded to genotype. 

### 2.3. Tissue Harvest and Sequencing

Mice were deeply anesthetized with isoflurane until unresponsive to tail pinch. Brains were rapidly removed and indicated tissue samples were obtained by tissue punch and frozen at −80 °C until use. Total RNA was isolated from mice brain tissue by using RNeasy Mini Kit (QIAGEN GmbH, Hilden, Germany) in accordance with the manufacturer’s protocols. The integrity of total RNA was validated by an Agilent bioanalyzer. A library was prepared from −1 ng RNA treated with the Ribo-Zero rRNA removal kit (Epicenter) per the manufacturer’s protocol. cDNA was blunt-ended, and an A base was added to the 3′ ends. Then Illumina sequencing adapters were ligated to the ends. Ligated fragments were amplified for 12 cycles using primers incorporating unique index tags. Fragments were sequenced on Illumina Hi-Seq-2500 or Hi-Seq-3000 using single reads extending 50 bases, targeting 25–30 million reads/sample.

RNA sequencing data were analyzed in the R software environment (https://www.r-project.org/, accessed on 17 March 2019) using the edgeR Bioconductor package to test for differential expression [26,27,28]. Pathway analysis was performed using Enrichr [29,30], and visualized with PathVisio [31,32] and Prism (GraphPad). qPCR data were also analyzed and visualized with Prism. Statistical significance was set at below 0.05, adjusting for multiple comparisons if necessary.

The mRNA expression of the genes of interest were validated by real-time qPCR. RNA was isolated from frozen dissected tissue and quantified using an ND1000 nanodrop spectrophotometer. cDNA was synthesized from 1 µg of RNA using a SuperScript II Reverse Transcriptase (Invitrogen, Thermo-Fisher Scientific, Waltham, MA) with random primers. Then, the product cDNA was mixed with PowerUP SYBR Green Master mix (Thermo-Fisher Scientific) and primers specific to the genes of interest were purchased from Bio-Rad (Hercules, CA, USA). Real-time PCR amplification was performed on a QuantStudio 3 system (Applied Biosystems, Foster City, CA, USA). Mouse GAPDH served as a comparator housekeeping gene. For GAPDH, the sequences used were *Fw: AGGTCGGTGTGAACGGATTTG* and *Rv: TGTAGACCATGTAGTTGAGGTCA*. The threshold cycle (Ct) value was defined as the cycle number at which the fluorescence crossed a fixed threshold above the baseline. For relative quantification, fold changes were measured using the ΔΔCt method. For each sample, the Ct value of mRNA was measured and compared with the GAPDH endogenous control as ΔCt (ΔCt = Ct_experiment_ − Ct_GAPDH_). The gene of interest mRNA fold change in the experimental sample relative to control samples was determined using 2^−ΔΔCt^ (ΔΔCt = ΔCt_experiment_ − ΔCt_control_). 

### 2.4. Measurement of 24S-HC in Hippocampus

Measurements were performed as previously described [10] from 58–128-day-old mice of both sexes. Briefly, 24S-HC in homogenized tissue was extracted with methanol. Deuterated 24-hydroxycholesterol-d_7_ (10 ng) was added as an internal standard. Extracted 24S-HC and the internal standard were derivatized with N,N-dimethylglycinate (DMG) to increase sensitivity. Oxysterol analysis was performed with a Shimadzu 20AD HPLC system and a LeapPAL autosampler coupled to a triple quadrupole mass spectrometer (API 4000) operating in MRM mode. The positive ion ESI mode was used for the detection of derivatized oxysterols. The study samples were injected in duplicate for data averaging. Data processing was conducted with Analyst 1.5.1 (Applied Biosystems). 24-HC data were normalized as ng/mg protein for various media.

## 3. Results

### 3.1. Behavioral Changes with Global and Partial Cyp46a1 Loss

Our work introduces a conditional knockout of *Cyp46a1* in most cortical and hippocampal excitatory neurons, using a VGLUT1-Cre driver line crossed with the *Cyp46a1*-floxed mice described in the Methods (cKO mouse). To demonstrate the overlap of cells expressing CYP46A1 with VGLUT1-expressing cells, Figure 1A shows RNA-seq data taken from the Allen Brain Atlas showing *Cyp46a1* expression in subsets of excitatory and inhibitory neurons from the cortex and hippocampus of mice, using the *Slc17a7* (VGLUT1) gene as a marker of glutamatergic neurons and the *Gad1* gene as a marker of GABAergic neurons. Figure 1B shows a schematic of the gene targeting strategy. 24S-HC levels in the hippocampus of cKOs (PND100–120) and KOs (PND100–550) are shown in Figure 1C, relative to matched-cohort WT mice. The effects of global knockouts were replicated from previous measures performed at PND 100 [14] to older animals (>PND400) here, as used in some behavioral and transcriptional studies below. Results were pooled given the absence of any notable difference from younger mice. These results show that most, but not all, 24-HC generation in the hippocampus can be attributed to excitatory neurons in mice. 

We next revisited the impact of *Cyp46a1* loss on behavior, including hippocampal dependent learning. KO mice demonstrated less overall ambulatory activity than WT controls. The ambulatory activity was not restored by a single systemic dose of SGE-301, a synthetic 24S-HC analog [11,34,35], administered immediately prior to assay (Figure 2A,B). Rearing frequency in KO mice was also lower than in WT controls, an effect that was partially mitigated by SGE-301 administration (Figure 2C) as evidenced by the increased levels in the KO mice and the lack of significant statistical difference between KO and WT within the treated group. Consistent with lower overall ambulatory activity, KO mice showed reduced travel in both the center and periphery of the chamber, and fewer entries into the maze center than controls, effects that were insensitive to SGE-301 treatment (Figure 2D–G). 

Interestingly, a trend toward hypolocomotion persisted in PND400 mice but did not reach statistical significance (Figure 3). The hypolocomotion phenotype was even less evident in conditional deletion mice at an age comparable to the global KO mice in Figure 2 (Figure 4). Thus, between PND400 global KO mice and conditional deletion mice, we have tools to investigate other behavioral phenotypes uncontaminated by the activity-related phenotype evident with *Cyp46a1* deletion at PND100. The results promise to aid the isolation of the effect of CYP46A1 on cognitive behavior, independent of the impact on locomotion. 

For hippocampal-dependent learning, we focused on contextual fear conditioning. KO mice (PND100) exhibited a peculiar phenotype in which baseline training, consisting of paired tone and shock, produced only a minor difference in freezing behavior from controls, evident during the final tone–shock pairing (Figure 5A), but subsequent contextual assessment revealed a consistently stronger freezing behavior (Figure 5). Increased freezing by KO mice was partly mitigated by the administration of SGE-301 prior to training on the previous day (Figure 5B). The re-presentation of the auditory cue in a novel context also produced stronger freezing (Figure 5C), but the effect was not discernibly affected by SGE-301 administration. 

To determine if the contextual-conditioned fear phenotype was affected by age, we tested a cohort of PND400 mice independent of those previously tested. Figure 6A–C shows that the essential phenotype persisted at this age in both context and auditory cue-based memory tests, at an age in which locomotor differences were not evident. 

To determine whether the selective, partial reduction of *Cyp46a1* could recapitulate these effects on contextual fear conditioning, we examined the effect of conditional deletion. Figure 6D–F shows that VGLUT1-selective deletion of *Cyp46a1* examined in PND100 animals failed to reproduce the characteristic contextual fear conditioning phenotype observed in Figure 5 and Figure 6A–C. The results suggest that selective loss of *Cyp46a1* and resulting partial reduction in 24S-HC (Figure 1) are insufficient to recapitulate the alterations in conditioning observed with global deletion. 

Using the global knockout mice (KOs), we also explored hippocampal-dependent spatial memory in the Morris Water Maze at an age at which conditioned fear deficits were robust (PND100). During acquisition, we found deficits in escape path length and escape latency for both cued (Figure 7A,B) and place (uncued) (Figure 7D,E) trials, with no differences in swim speeds (Figure 7C,F). This pattern of results suggests that there was no hypolocomotion (measured by swim speed), which could explain the differences between genotypes in task acquisition. The cued performance deficits suggest possible visual, sensorimotor, or emotionality alterations, especially early on in the trials. However, at the end of both the cued and place training, the KO mice were performing at levels similar to controls and went on to exhibit no differences in spatial memory retention, measured by time spent in the target quadrant 1 h and 48 h post-acquisition during the probe trials (Figure 7G,H). These results contrast with previously published results on PND90–120 *Cyp46a1* deficient mice [13] and are addressed further in the Discussion. 

### 3.2. Transcriptional Changes with Complete and Selective Cyp46a1 Loss

Behavioral changes could be associated with the acute loss of 24S-HC but could also be dependent on downstream changes in other gene products. To assay for transcriptional changes, we performed bulk differential RNA-seq analysis of the hippocampus at different ages, comparing global *Cyp46a1*^−/−^ KO mice with WT. Summary results are shown in Figure 8, with details of top gene alterations given in Table 1, Table 2, Table 3, Table 4 and Table 5. Changes in cholesterol biosynthetic pathways were over-represented, as outlined in Appendix A. These changes may be homeostatic given that cholesterol levels have not been demonstrated to be altered in *Cyp46a1*-deficient animals [18,19]. *Abca1* and *Apoe* are known targets of LXR, an oxysterol-dependent transcriptional regulator [15,20,36], and these targets were upregulated in our study, a direction opposite of that expected of an LXR agonist role for 24S-HC. 

We chose several transcripts common to CA1 and the dentate gyrus, including the LXR targets, for qPCR validation (Figure 9). Results of these analyses for both global deletion and selective deletion in excitatory neurons are given in Figure 9. At PND100, results clearly followed the RNA-seq results (Figure 9A). In cKO mice, however, the results failed to replicate, except for 2 transcripts, *Fdft1* and *Abca1* (Figure 9B). At an older age, transcriptional changes in mice with global deletion persisted, albeit at smaller effect sizes for some transcripts (Figure 9C). Overall, the results validate findings from RNA-seq analysis but suggest that transcriptional changes are dramatically decreased with the selective deletion of *Cyp46a1*. 

## 4. Discussion

Our study explores the role of cholesterol metabolism in the brain, with a focus on the enzyme CYP46A1 and its oxidation product, 24S-HC. Cholesterol metabolism pathways have garnered significant attention in the context of brain disorders, such as neurodegenerative illnesses and neuropsychiatric disorders [1,4,5,6]. Although previous research has established the importance of CYP46A1 and 24S-HC in various aspects of brain function [7,17,37,38], there are still many unanswered questions and discrepancies in the literature. For instance, *Cyp46a1* overexpression has been associated with benefits in models of Huntington’s disease, spinal cerebellar ataxia, and Alzheimer’s disease [39,40,41,42,43,44] but may be detrimental in excitotoxicity models [5,10]. *Cyp46a1* loss through acute knockdown has also been associated with lipid dysregulation, different than the effects of chronic loss, where cholesterol levels are unaffected [18,45]. Our results demonstrate new behavioral phenotypes and introduce a conditional knockout approach that reduces 24S-HC levels and yields intermediate behavioral and transcriptional effects. We suggest that the dose effects of 24S-HC likely explain differences in behavioral and transcriptional phenotypes between the KO and cKO mice. A simplified summary of our major results is shown in Appendix A. 

Our findings provide insights into the impact of the conditional knockout of *Cyp46a1* in most forebrain excitatory neurons, compared with the impact of global deletion. Previous work has shown that 24S-HC levels increase with development to near-adult levels by PND30 [8]. We observed a substantial reduction in 24S-HC levels in the hippocampus of VGLUT1-driven conditional deletion in adult mice (Figure 1C), confirming the contribution of excitatory neurons to 24S-HC production in the brain. These results align with previous expression studies [21], supporting the idea that CYP46A1 in excitatory neurons plays a key role in cholesterol metabolism in the hippocampus. The results suggest that other cell types, likely GABAergic interneurons (Figure 1), also have a substantial role that is sufficient to maintain WT behavior and transcriptional levels. We note that our measurements were focused on the hippocampus because of our initial expectation of spatial memory deficits [13], and expression results from the Allen Brain Atlas (Figure 1) are focused on the hippocampus and cortex. Given the results of contextual fear conditioning, amygdala levels of 24S-HC will be of interest to explore in future studies. Past work has demonstrated similar levels of CP46A1 and 24S-HC in the amygdala compared with the hippocampus [8,9,46], and most cells of the amygdala are VGLUT1 positive [47], suggesting that 24S-HC levels in the amygdala should be reduced in cKO mice. We also acknowledge that deletion in a subset of cells does not guarantee a loss of function in those cells, since 24S-HC is secreted and, thus, can potentially have non-cell autonomous effects.

Our study also investigated the behavioral consequences of *Cyp46a1* loss. Global deletion but not conditional deletion decreased ambulatory activity at PND100, lowered rearing frequency, and reduced exploration in the center and periphery of the chamber. These behavioral changes suggest a potential role for CYP46A1 in regulating locomotion and possibly other functions that were not previously identified [13]. Interestingly, the hypolocomotion phenotype was not as strong at PND400. The less pronounced hypolocomotion phenotype in PND400 knockouts suggests that age influences the impact of *Cyp46a1* deletion on behavior and parallels some transcriptional effects (Figure 9).

For hippocampal-dependent learning, we tested contextual fear conditioning and spatial learning in a Morris Water Maze. In fear conditioning, knockout mice displayed a unique phenotype, with minor differences in freezing behavior during baseline training but consistently stronger freezing behavior during subsequent contextual assessments. These results could suggest better memory retention with a loss of 24S-HC, a potentially beneficial effect of 24S-HC loss but opposite of that predicted by the positive allosteric pharmacological effects of 24S-HC on NMDAR function [11,14]. The hypolocomotion phenotype, particularly in the mice with global deletion at PND100, could partially contribute to the large differences in freezing behavior. However, no differences in freezing were observed during the baseline period before the tone/shock presentation, indicating that similar levels of baseline freezing were present in this test. We do not suspect that the perception of the shock stimulus contributed to this difference in freezing behavior, as both genotypes show similar thresholds for flinching behavior in the shock sensitivity tests. This suggests a complex role for CYP46A1 in memory formation and consolidation. Importantly, the administration of SGE-301 prior to acquisition partially mitigated the increased freezing behavior measure on subsequent days, suggesting a direct role for acute 24S-HC at the time of acquisition in the fear memory phenotype.

To further understand the effects of *Cyp46a1* loss on cognitive function, we explored hippocampal-dependent spatial memory in PND100 mice. Deficits in escape path length and escape latency during both cued and place trials were observed. Importantly, swim speed did not explain these differences, suggesting that hypolocomotion is not the primary factor contributing to task acquisition deficits. However, towards the end of place acquisition, knockout mice performed similarly to controls and exhibited no differences in spatial memory retention, during probe trials implying that *Cyp46a1* deletion primarily affects early task acquisition. These results are dramatically different than those observed by previous investigations of *Cyp46a1* global deletion [13] and may hint at the importance of background strain on outcomes. 

To uncover potential mechanisms underlying the observed behavioral alterations, we conducted RNA-seq analysis of the hippocampus at different ages. Although we validated major RNA-seq results with qPCR analysis, this work remains hypothesis-generating because we cannot trace transcriptional changes directly to behavioral observations. Nevertheless, changes in cholesterol biosynthetic pathways were over-represented in transcriptional changes, with some unexpected regulatory patterns of LXR target genes, such as *Abca1* and *Apoe*. The validation of selected transcripts through qPCR revealed that most of the results aligned with the RNA-seq findings. LXR targets *Apoe* and *Abca1* were upregulated by *Cyp46a1* loss. This change is unexpected based on the canonical understanding of the interaction of oxysterols with LXR, but our observations are consistent with a previous study [17]. Despite the paradoxical nature of the change, the upregulation of *Abca1* was recently found to ameliorate neurodegeneration and neuroinflammation in a tauopathy model, suggesting potential beneficial effects of 24S-HC reduction [48]. The selective deletion of *Cyp46a1* in excitatory neurons substantially reduced the transcriptional changes, suggesting a dose-dependent relationship between CYP46A1, 24S-HC, and gene expression.

Our results contribute to the growing body of knowledge regarding the role of CYP46A1 and 24S-HC in brain function. They underscore the complexity of this system, with effects on locomotion, contextual/emotional memory, and gene expression observed especially in global knockouts. We have previously suggested that CYP46A1 reduction may be therapeutic in certain situations [5]. The present results suggest that a nearly complete reduction in enzymatic function may be needed for functional effects, since partial genetic loss was insufficient to recapitulate key results. Future research should further explore the intricate transcriptional regulation by CYP46A1 and the mechanisms underlying the observed behavioral alterations. Our results also do not preclude signaling functions for CYP46A1/24S-HC beyond LXR-mediated transcriptional regulation and neuromodulation.

In conclusion, our study provides a deeper understanding of the impact of *Cyp46a1* deletion on brain function and offers new insights into the potential behavioral significance of targeting cholesterol metabolism pathways. Our work shows that under certain circumstances, *Cyp46a1* deletion can have significant effects on locomotion, learning/conditioning, and emotionality, and the cued trials results of the Morris Water Maze suggest that other functions are also affected that could impact behavioral indices. Thus, the multifaceted nature of these effects warrants the further investigation of the underlying mechanisms and the development of targeted interventions.

## Figures and Tables

**Figure 1 biomolecules-14-00254-f001:**
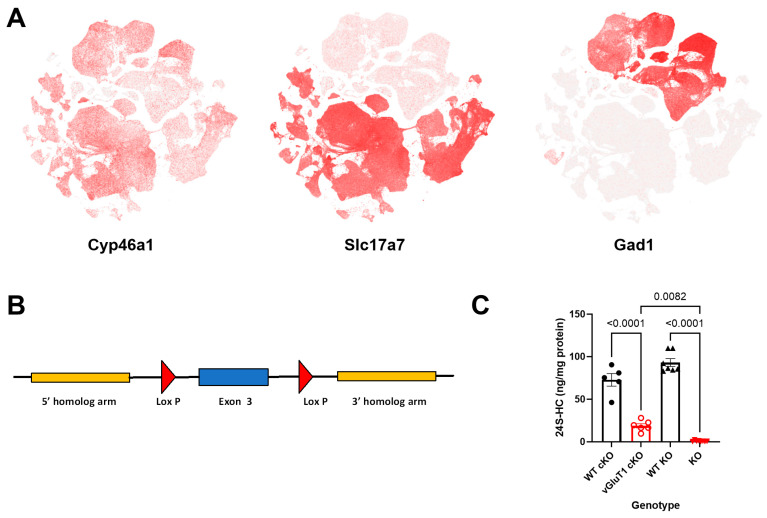
Impact of constitutive and conditional deletion of *Cyp46a1* on 24S-HC levels in hippocampus. (**A**) Expression of *Cyp46a1* transcripts in excitatory (*Slc17a7* overlap) and inhibitory (*Gad1* overlap) neurons in hippocampus and cortex from Allen Brain Atlas 10× transcriptomics database [33]. (**B**) Schematic depicting the strategy for conditional deletion. (**C**) 24S-HC levels measured in hippocampus of the indicated genotypes. WT cKO and WT KO refer to WT animals, matched and tested simultaneously with the respective deletion mice. The results of a one-way ANOVA showed a significant effect of genotype on 24S-HC levels (F = 136.6, *p* < 0.0001). Results of Tukey’s multiple comparison tests are shown with *p* values indicated above the comparison bars.

**Figure 2 biomolecules-14-00254-f002:**
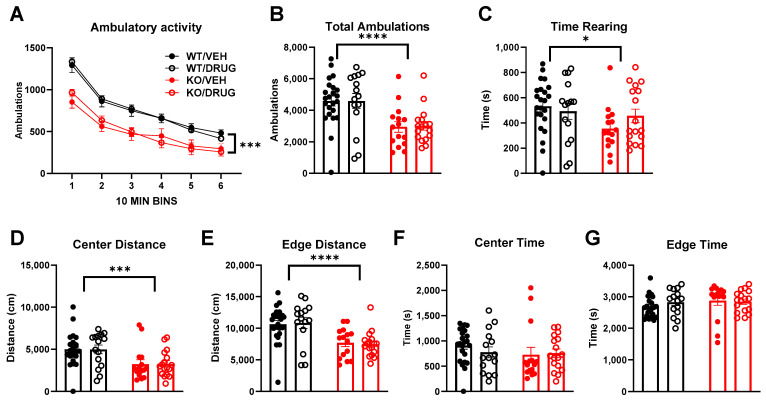
Impact of global *Cyp46a1* deletion (KO) and SGE-301 (20 mg/kg, drug) on ambulatory behaviors in mice. (**A**) Total ambulations by time block as impacted by genotype and drug. A three-way ANOVA showed a reduction in locomotor behavior with genotype (*p* < 0.0001, F (1, 66) = 19.93) and time (*p* < 0.0001, F (3.886, 256.5) = 177.5) but no effect of drug F (1, 66) = 0.006688) and no interactions. (**B**) Summary of total cumulative ambulations over 1 h, showing an effect of genotype but not an effect of drug (two-way ANOVA, *p* < 0.0001, F (1, 66) = 19.93). (**C**) Time spent rearing showed an effect of genotype (two-way ANOVA, *p* = 0.04, F (1, 66) = 4.199), post hoc comparisons show vehicle-treated WT mice rearing significantly more than vehicle-treated KO mice (*p* = 0.03) while the difference between genotypes in the drug-treated mice was not significant (*p* > 0.99). (**D**,**E**) Distance traveled in center and edge areas respectively (*p* < 0.001, F (1, 66) = 14.41 and F (1, 66) = 22.62 respectively). (**F**,**G**) Time spent in center and edge areas respectively. None of the two-way ANOVAs revealed an effect of drug. Asterisks indicate * *p* < 0.05, *** *p* < 0.0001, **** *p* < 0.00001.

**Figure 3 biomolecules-14-00254-f003:**
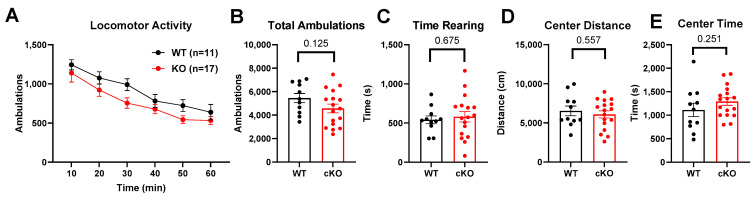
Ambulatory behavior in PND400 mice. (**A**) A two-way ANOVA for ambulatory activity revealed an effect of time but no effect of genotype and no interaction between genotype and time. (**B**–**E**) *p* values from unpaired *t*-tests are indicated above bars in (**B**–**E**) for cumulative behaviors.

**Figure 4 biomolecules-14-00254-f004:**
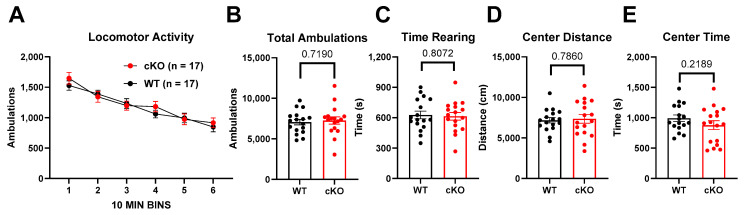
Effect of selective deletion of *Cyp46a1* in excitatory neurons (cKO) on locomotor behaviors in PND100 mice. (**A**) A two-way ANOVA for ambulatory activity revealed an effect of time but no main effect of genotype and no interaction between genotype and time. (**B**–**E**) *p* values for unpaired *t*-tests are shown above the graphs for the indicated cumulative behaviors.

**Figure 5 biomolecules-14-00254-f005:**
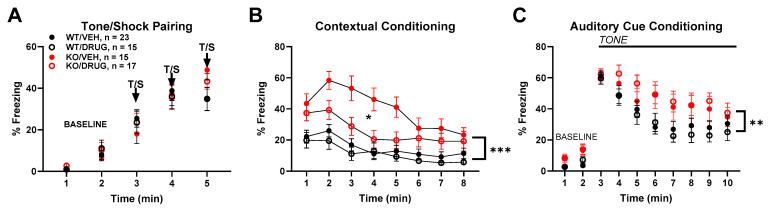
Effect of *Cyp46a1*^−/−^ genotype (KO) and SGE-301 (20 mg/kg, drug) on conditioned fear learning at PND100. Drug was administered 2 h prior to day 1 tone–shock pairing and not re-administered for conditioning probes on subsequent days shown in panels (**B**,**C**). (**A**) Effect of tone–shock (T/S) pairing on freezing behavior in conditioned fear training. A three-way ANOVA revealed a main effect of time (F (2.825, 186.5) = 108.3, *p* < 0.0001) and a time by genotype interaction (F (4, 264) = 3.159, *p* = 0.0147) but no main effect of genotype (F (1, 66) = 0.1629, *p* = 0.6878). (**B**) Freezing behaviors induced on day 2 with the return of the animal to the context of the tone–shock pairing. A three-way ANOVA revealed main effects of time (F (3.775, 249.2) = 20.25, *p* < 0.0001), genotype (F (1, 66) = 28.02, *p* < 0.0001)), and drug (F (1, 66) = 5.949, *p* = 0.0174). In addition, there were interactions between time and genotype (F (7, 462) = 2.148, *p* = 0.0376) and a three-way interaction between time, genotype, and drug (F (7, 462) = 2.118, *p* = 0.0404). (**C**) For auditory cue testing on day 3, a three-way ANOVA revealed main effects of time (F (2.084, 137.5) = 86.30, *p* < 0.0001) and genotype (F (1, 66) = 9.154, *p* = 0.0035), as well as a time by genotype interaction (F (9, 594) = 2.629, *p* = 0.0055). Asterisks: * *p* < 0.05, ** *p* < 0.01, *** *p* < 0.0001.

**Figure 6 biomolecules-14-00254-f006:**
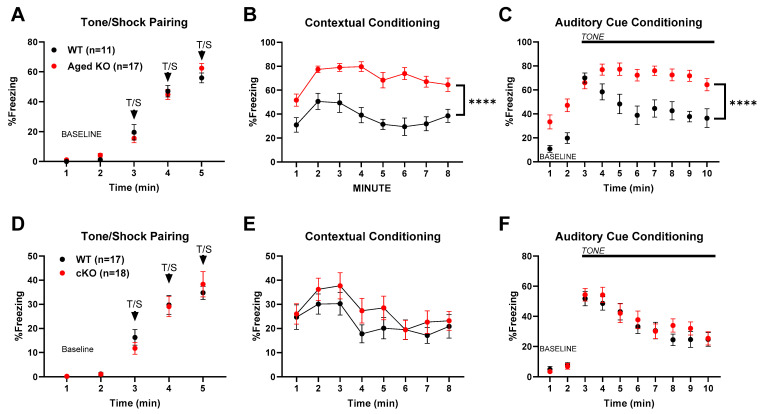
Contextual fear conditioning phenotype is retained in PND400 *Cyp46a1*^−/−^ mice but not VGLUT1-Cre *Cyp46a1* cKO mice. (**A**–**C**) PND 400 WT and *Cyp46a1*^−/−^ mice. (**A**) A two-way ANOVA revealed only a main effect of time (F (3.051, 79.33) = 208.2, *p* < 0.0001) but no main effect of genotype (F (1, 26) = 0.1259, *p* = 0.7256) or genotype by time interaction (F (4, 104) = 1.309, *p* = 0.2717). (**B**) Testing of freezing on day 2 in the context of the previous tone–shock pairing showed a main effect of time (F (4.995, 129.9) = 7.150, *p* < 0.0001) and genotype (F (1, 26) = 35.31, *p* < 0.0001). (**C**) Tone presentation on day 3 showed a main effect of time (F (2.712, 70.52) = 25.66, *p* < 0.0001), genotype (F (1, 26) = 18.40, *p* = 0.0002), and an interaction between time and genotype (F (9, 234) = 3.725, *p* = 0.0002). (**D**–**F**) Phenotype of conditional *Cyp46a1* loss in VGLUT1-Cre mice. (**D**) No effect of genotype on acquisition of freezing behavior (F (1, 33) = 0.01925, *p* = 0.8905). (**E**) No effect of conditional *Cyp46a1* loss on context dependent freezing F (1, 33) = 1.246, *p* = 0.2723). (**F**) No effect of conditional loss on tone-dependent freezing (F (1, 33) = 0.3456, *p* = 0.5606). Asterisks: **** *p* < 0.00001.

**Figure 7 biomolecules-14-00254-f007:**
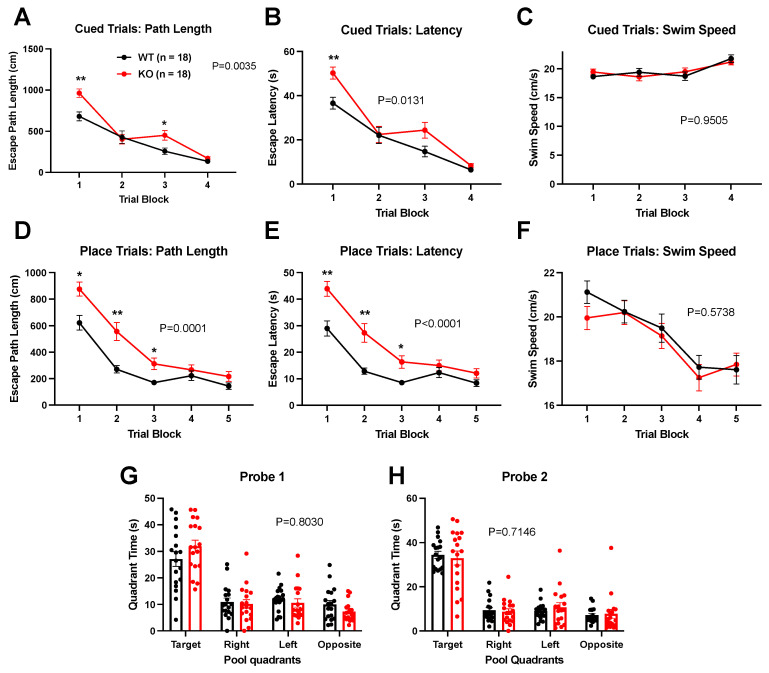
Effect of *Cyp46a1*^−/−^ genotype on the acquisition and retention of spatial learning in Morris Water Maze at PND100. For all panels, *p* values show the values for the main effect of genotype for each component. (**A**) There was a main effect of genotype (F (1, 34) = 9.867, *p* = 0.0035) and an interaction between trial and genotype (F (3, 102) = 4.249, *p* = 0.0072), and post hoc comparisons show that KO mice were significantly different from WT during blocks 1 (** *p* = 0.0024) and 3 (* *p* = 0.037). (**B**) There was a main effect of genotype (F (1, 34) = 6.849, *p* = 0.0131) and an interaction between genotype and trial (F (3, 102) = 3.153, *p* = 0.0281), and post hoc comparisons show that KO mice were significantly different from WT during block 1 (** *p* = 0.0047). (**C**) There was no effect of genotype on cued trial swim speed (F (1, 34) = 0.003904, *p* = 0.9505). (**D**) There was a main effect of genotype (F (1, 34) = 19.37, *p* = 0.0001) and interaction between genotype and trial (F (4, 136) = 3.957, *p* = 0.0045) on path length to escape. Post hoc comparisons show that KOs were significantly different from WT during blocks 1 (* *p* = 0.0108), 2 (** *p* = 0.0036), and 3 (* *p* = 0.034). (**E**) Latency to escape during place trials showed a main effect of genotype (F (3.361, 114.3) = 59.60, *p* < 0.0001) and interaction between genotype and trial (F (4, 136) = 4.419, *p* = 0.0022), and post hoc comparisons again show that KOs were significantly different from WT during blocks 1 (** *p* = 0.0036), 2 (** *p* = 0.0042), and 3 (* *p* = 0.023). (**F**) Swim speed during place trials showed no main effect of genotype (F (1, 34) = 0.3226, *p* = 0.5738). (**G**,**H**). Tests of time in quadrants of the pool with platform removed were performed 1 h (Probe 1) and 48 h (Probe 2) following the final training session. No differences between WT and KO animals in time in target quadrant were revealed. P value for the effect of genotype in a two-way ANOVA is given above the graphs.

**Figure 8 biomolecules-14-00254-f008:**
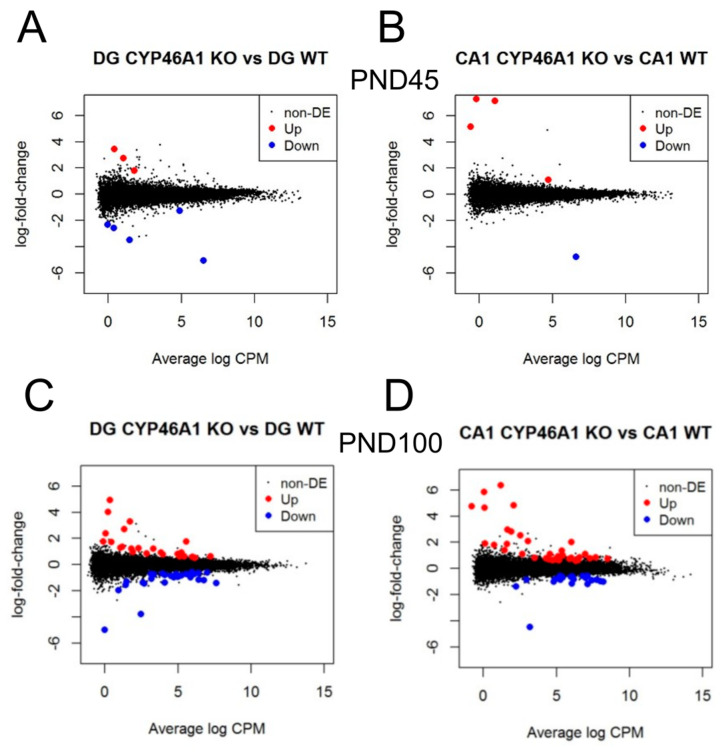
Differential transcription between *Cyp46a1*^−/−^ (CYP46A1 KO) and WT controls in dentate gyrus (DG; (**A**,**C**)) and CA1 (**B**,**D**) hippocampi. (**A**,**B**) Results from postnatal day 45 mice; (**C**,**D**) results from postnatal day 100 mice. *n* = 3 mice of each genotype. The graphs show log-fold change between genotypes as a function of the average counts per million (CPM). Black dots show non-significant changes. Red dots show increased expression, and blue dots show decreased expression (false discovery rate < 0.05). The lowest blue point in each graph is *Cyp46a1*. Details are given in Table 1, Table 2, Table 3 and Table 4.

**Figure 9 biomolecules-14-00254-f009:**
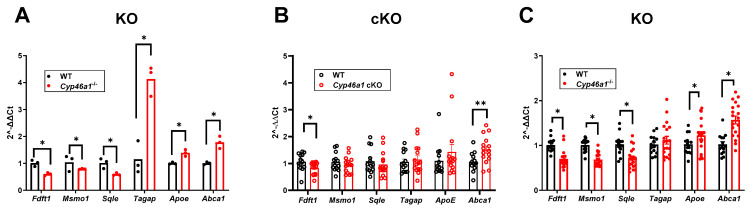
qPCR validation of RNA-seq results in PND100 animals (**A**,**B**) and PND400 animals ©. Statistical differences are indicated from unpaired *t* test analysis. Impacts of global loss (*Cyp46a1^−/−^*; KO) in PND100 and PND400 only partly replicated in conditional knock-out (cKO) PND100 mice. * *p* < 0.05, ** *p* < 0.001.

**Table 1 biomolecules-14-00254-t001:** Differentially expressed genes in the CA1 of PND45 animals.

Gene	LogFC	LogCPM	*p* Value	FDR	Gene Name
*Cyp46a1*	−4.749	6.600	1.78 × 10^−141^	2.58 × 10^−137^	cytochrome P450, family 46, subfamily a, polypeptide 1
*Gm8420*	7.112	1.062	1.88 × 10^−22^	1.36 × 10^−18^	predicted gene 8420
*Gm14094*	7.278	−0.200	6.78 × 10^−13^	3.27 × 10^−9^	predicted gene 14094
*Olfr31*	5.142	−0.603	1.57 × 10^−8^	5.65 × 10^−5^	olfactory receptor 31
*Rbm43*	1.109	4.735	2.53 × 10^−6^	0.007305	RNA binding motif protein 43

**Table 2 biomolecules-14-00254-t002:** Differentially expressed genes in the dentate gyrus of PND45 animals.

Gene	LogFC	LogCPM	*p* Value	FDR	Gene Name
*Cyp46a1*	−6.040	6.527	5.55 × 10^−112^	8.19 × 10^−108^	cytochrome P450, family 46, subfamily a, polypeptide 1
*Trim21*	3.453	0.427	2.10 × 10^−7 ^	0.001548	tripartite motif containing 21
*4930481B07Rik*	2.744	1.040	1.89 × 10^−6^	0.009277	4930481B07Rik RIKEN cDNA
*Chst3*	−2.570	0.367	3.67 × 10^−6^	0.013563	carbohydrate sulfotransferase 3
*Insig1*	−1.260	4.886	8.31 × 10^−6^	0.024536	insulin induced gene 1
*Hhe×*	−2.333	−0.026	1.32 × 10^−5^	0.032547	hematopoietically expressed homeobox
*Tox3*	1.811	1.779	2.03 × 10^−5^	0.038145	tox high mobility group box family member 3
*Otos*	−3.473	1.457	2.07 × 10^−5^	0.038145	otospiralin

**Table 3 biomolecules-14-00254-t003:** Differentially expressed genes in the CA1 of PND100 animals.

Gene	LogFC	LogCPM	*p* Value	FDR	Gene Name
*Cyp46a1*	−4.472	3.202	1.15 × 10^−47^	1.64 × 10^−43^	cytochrome P450, family 46, subfamily a, polypeptide 1
*Gm8730*	4.835	2.075	8.31 × 10^−33^	5.93 × 10^−29^	predicted pseudogene 8730
*Gm8420*	6.367	1.185	4.22 × 10^−31^	2.01 × 10^−27^	predicted gene 8420
*Tagap*	2.952	1.641	3.75 × 10^−18^	1.34 × 10^−14^	T-cell activation Rho GTPase-activating protein
*Tmem181b.ps*	2.101	3.050	1.72 × 10^−17^	4.91 × 10^−14^	transmembrane protein 181B, pseudogene
*Gm9625*	2.509	2.528	2.20 × 10^−17^	5.22 × 10^−14^	predicted gene 9625
*Ccr6*	5.854	0.071	1.05 × 10^−15^	2.14 × 10^−12^	Mus musculus chemokine (C-C motif) receptor 6
*Gm14094*	4.628	0.098	6.65 × 10^−15^	1.19 × 10^−11^	predicted gene 14094
*Ubc*	−1.025	8.212	4.82 × 10^−14^	7.64 × 10^−11^	ubiquitin C
*Fdft1*	−1.172	6.065	2.96 × 10^−11^	4.22 × 10^−8^	Squalene synthase
*Msmo1*	−1.188	7.134	8.37 × 10^−10^	1.09 × 10^−6^	Methylsterol monooxygenase 1
*Rplp0*	−0.909	7.425	2.68 × 10^−9^	3.18 × 10^−6^	ribosomal protein, large, P0
*Rplp0.ps1*	4.744	−0.799	8.43 × 10^−9^	8.94 × 10^−6^	ribosomal protein, large, P0, pseudogene 1
*Nsdhl*	−0.868	5.274	8.77 × 10^−9^	8.94 × 10^−6^	Sterol-4-alpha-carboxylate 3-dehydrogenase, decarboxylating
*Gm29340*	2.834	1.906	1.18 × 10^−7^	0.00011183	predicted gene 29340
*Sqle*	−1.012	4.827	4.30 × 10^−7^	0.000383502	squalene epoxidase
*Ptprg*	1.003	5.433	1.09 × 10^−6^	0.000912133	receptor-type tyrosine-protein phosphatase gamma isoform 2
*Hipk1*	1.091	6.961	1.73 × 10^−6^	0.001368802	Homeodomain-interacting protein kinase 1
*Fdps*	−0.758	5.250	1.87 × 10^−6^	0.001387266	Farnesyl pyrophosphate synthase
*Dbi*	−0.861	7.751	1.94 × 10^−6^	0.001387266	diazepam binding inhibitor
*Ly86*	−0.828	4.908	4.14 × 10^−6^	0.002783167	Lymphocyte antigen 86
*Atf4*	−0.818	6.185	4.29 × 10^−6^	0.002783167	Cyclic AMP-dependent transcription factor ATF-4
*Asb1*	2.003	6.010	5.17 × 10^−6^	0.003108183	Ankyrin repeat and SOCS box protein 1
*Fn1*	1.088	4.560	5.23 × 10^−6^	0.003108183	Fibronectin Anastellin
*Arhgef37*	1.913	0.135	6.31 × 10^−6^	0.003604708	Rho guanine nucleotide exchange factor 37
*Sc5d*	−0.735	6.988	8.22 × 10^−6^	0.004513714	sterol-C5-desaturase
*Gm7180*	1.790	0.740	1.16 × 10^−5^	0.006143589	predicted pseudogene 7180
*Clasp1*	0.822	6.013	1.25 × 10^−5^	0.00638559	Mus musculus CLIP associating protein 1
*Htr1a*	0.603	6.326	1.31 × 10^−5^	0.006427525	5-hydroxytryptamine (serotonin) receptor 1A
*Ilf3*	0.774	7.299	2.96 × 10^−5^	0.014060774	Interleukin enhancer-binding factor 3
*Abca1*	0.926	5.119	3.96 × 10^−5^	0.01825061	ATP-binding cassette, sub-family A member 1

**Table 4 biomolecules-14-00254-t004:** Differentially expressed genes in the dentate gyrus of PND100 animals.

Gene	LogFC	LogCPM	*p* Value	FDR	Gene Name
*Cyp46a1*	−3.770	2.482	5.07 × 10^−36^	7.49 × 10^−32^	cytochrome P450, family 46, subfamily a, polypeptide 1
*Ccr6*	4.937	0.363	2.82 × 10^−18^	2.08 × 10^−14^	Mus musculus chemokine (C-C motif) receptor 6
*Hmgcs1*	−1.423	7.614	2.51 × 10^−15^	1.24 × 10^−11^	Hydroxymethylglutaryl-CoA synthase, cytoplasmic
*Tagap*	2.713	1.343	1.45 × 10^−14^	5.34 × 10^−11^	T-cell activation Rho GTPase-activating protein
*Ldlr*	−1.370	4.099	6.47 × 10^−14^	1.72 × 10^−10^	Low-density lipoprotein receptor
*Msmo1*	−1.184	6.772	6.98 × 10^−14^	1.72 × 10^−10^	Methylsterol monooxygenase 1
*Scd1*	−1.156	6.407	4.11 × 10^−12^	8.67 × 10^−9^	stearoyl-Coenzyme A desaturase 1
*1700023H06Rik*	4.020	0.221	2.71 × 10^−11^	5.00 × 10^−8^	RIKEN cDNA 1700023H06 gene
*Aplnr*	−4.997	−0.024	3.06 × 10^−10^	5.01 × 10^−7^	Apelin receptor
*Foxn3*	−0.731	6.114	1.88 × 10^−8^	2.66 × 10^−5^	Forkhead box protein N3
*Fdft1*	−0.978	5.866	1.98 × 10^−8^	2.66 × 10^−5^	Squalene synthase
*Gm29340*	3.305	1.711	2.24 × 10^−7^	0.000275929	predicted gene 29340
*Rgs13*	−1.373	2.612	3.93 × 10^−7^	0.000431055	Regulator of G-protein signaling 13
*Slc27a3*	1.245	2.314	4.09 × 10^−7^	0.000431055	solute carrier family 27, member 3
*Erap1*	−1.083	3.175	5.77 × 10^−7^	0.000568015	Endoplasmic reticulum aminopeptidase 1
*Sqle*	−0.984	4.735	1.50 × 10^−6^	0.001384955	squalene epoxidase
*Lrrc55*	−0.867	4.527	2.35 × 10^−6^	0.002044269	Leucine-rich repeat-containing protein 55
*Pik3r1*	−0.635	6.415	3.26 × 10^−6^	0.002674551	phosphoinositide-3-kinase regulatory subunit 1
*Acvr1c*	−0.887	5.867	4.81 × 10^−6^	0.003737493	Activin receptor type-1C
*Slc7a11*	−0.948	4.993	7.18 × 10^−6^	0.005301361	solute carrier family 7, member 11
*Nhlh1*	−1.539	1.424	8.16 × 10^−6^	0.005740722	nescient helix loop helix 1
*Prkd3*	−0.780	4.093	1.28 × 10^−5^	0.008590708	protein kinase D3
*Ddn*	0.626	7.228	1.99 × 10^−5^	0.012805329	Dendrin
*Icam1*	−1.971	0.929	2.27 × 10^−5^	0.013449952	intercellular adhesion molecule 1
*Fst*	1.216	3.318	2.28 × 10^−5^	0.013449952	follistatin FST315
*Idh1*	−0.569	5.968	2.51 × 10^−5^	0.014240691	Isocitrate dehydrogenase [NADP] cytoplasmic
*Arsi*	1.359	1.213	2.70 × 10^−5^	0.014767094	Arylsulfatase I
*Ube2t*	1.207	1.824	2.98 × 10^−5^	0.015715329	ubiquitin-conjugating enzyme E2T
*Insig1*	−0.753	3.350	3.28 × 10^−5^	0.01672288	Insulin-induced 1 protein

**Table 5 biomolecules-14-00254-t005:** Differentially expressed genes in the CA1 of PND400 animals.

Gene	LogFC	LogCPM	*p* Value	FDR	Gene Name
*Cyp46a1*	−4.283	5.941	9.79 × 10^−52^	1.47 × 10^−47^	cytochrome P450, family 46, subfamily a, polypeptide 1
*Msmo1*	−1.382	6.848	1.81 × 10^−20^	1.37 × 10^−16^	methylsterol monoxygenase 1
*Fdft1*	−1.457	5.659	5.61 × 10^−18^	2.81 × 10^−14^	farnesyl diphosphate farnesyl transferase 1
*Sqle*	−1.515	5.548	5.37 × 10^−16^	2.02 × 10^−12^	squalene epoxidase
*Hmgcs1*	−1.24	7.655	1.35 × 10^−14^	4.06 × 10^−11^	3-hydroxy-3-methylglutaryl-Coenzyme A synthase 1
*Cyp51*	−1.082	5.776	9.99 × 10^−13^	2.50 × 10^−9^	cytochrome P450, family 51
*Hmgcr*	−1.099	6.661	1.86 × 10^−11^	4.00 × 10^−8^	3-hydroxy-3-methylglutaryl-Coenzyme A reductase
*Insig1*	−1.711	5.536	4.44 × 10^−10^	8.35 × 10^−7^	insulin induced gene 1
*Dhcr24*	−1.354	4.014	2.49 × 10^−9^	4.17 × 10^−6^	24-dehydrocholesterol reductase
*Ldlr*	−2.279	3.86	6.37 × 10^−8^	9.59 × 10^−5^	low density lipoprotein receptor
*Cdr1*	−0.743	6.427	7.39 × 10^−8^	0.000101	cerebellar degeneration related antigen 1
*Nsdhl*	−1.047	4.737	2.70 × 10^−7^	0.000338	NAD(P) dependent steroid dehydrogenase-like
*Abca1*	1.049	5.172	3.38 × 10^−7^	0.000391	ATP-binding cassette, sub-family A (ABC1), member 1
*Adarb1*	−0.992	5.413	3.98 × 10^−7^	0.000421	adenosine deaminase, RNA-specific, B1
*Rgs4*	−1.128	7.309	4.24 × 10^−7^	0.000421	regulator of G-protein signaling 4
*Sostdc1*	−6.799	3.501	4.48 × 10^−7^	0.000421	sclerostin domain containing 1
*Ccbe1*	−5.451	1.559	1.28 × 10^−6^	0.001129	collagen and calcium binding EGF domains 1
*Mvd*	−0.979	4.083	2.25 × 10^−6^	0.001871	mevalonate (diphospho) decarboxylase
*Sh2d3c*	−1.142	3.226	2.36 × 10^−6^	0.001871	SH2 domain containing 3C
*Sc5d*	−0.675	6.354	2.75 × 10^−6^	0.001975	sterol-C5-desaturase
*Dnajc6*	−0.626	6.977	2.76 × 10^−6^	0.001975	DnaJ heat shock protein family (Hsp40) member C6
*Sv2b*	−0.658	8.158	2.91 × 10^−6^	0.00199	synaptic vesicle glycoprotein 2 b
*Gm10384*	2.935	−0.252	4.33 × 10^−6^	0.002834	predicted gene 10384
*Npr3*	−1.129	3.34	5.82 × 10^−6^	0.003652	natriuretic peptide receptor 3
*Kcng2*	−1.532	2.191	6.82 × 10^−6^	0.004102	potassium voltage-gated channel, subfamily G, member 2
*C4b*	1.041	5.116	7.12 × 10^−6^	0.004123	complement component 4B (Chido blood group)
*Cfap45*	−2.203	1.157	8.71 × 10^−6^	0.004855	cilia and flagella associated protein 45
*Pcp4*	−3.375	4.502	1.21 × 10^−5^	0.006514	Purkinje cell protein 4
*Nppc*	−1.673	3.004	1.28 × 10^−5^	0.006631	natriuretic peptide type C

## Data Availability

Raw data are available from the authors on reasonable request.

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
