# Peer review of "Effects of Complete and Partial Loss of the 24S-Hydroxycholesterol-Generating Enzyme Cyp46a1 on Behavior and Hippocampal Transcription in Mouse"

_biomolecules, 2024, doi:10.3390/biom14030254_

Round 1

Reviewer 1 Report

Comments and Suggestions for Authors

The Authors analyzed the behavioral effects of 24-hydroxycholesterol (24S-HC) loss in mice carrying a constitutive deletion of the Cyp46A1 gene and in a new conditional mouse line, also focusing on the transcriptional modulation of cholesterol metabolism genes.

Despite the relevance of the topic, the manuscript has numerous shortcomings and for this reason, in this version, is not suitable for publication.

1) Experimental model design: it is not clear which are the differences between the two models used (e.g.  in conditional Cyp46A1 KO mice are the excitatory neurons the only ones suffering for Cyp46A1? What do the Authors mean with the term “whole-body Cyp46A1 deletion”?). The Authors have to describe the most important features of the models and explain why they have chosen and compared them. In addition, why PDN400 analyses have been performed only in constitutive KO mice and why SGE-301 has been employed only in ambulatory behavior and learning studies?

2) Results are not clear: in particular the different groups are not properly named; e.g. in Fig 1C and Fig 6, X axis labels, and in discussion. Which mouse models do the Authors mean by WTcKO, vGluT1 cKO, WT KO, KO, and GABAergic interneurons? The same convention must be used throughout the text for identifying the different groups and explained the first time.  This is mandatory to understand data. Data on transcriptional changes need more in-depth investigation since in the present version are not informative.  

3) Statistic analysis is missing for Fig 1C, while in Fig 9 statistic comparison had to be made also between KO and cKO mice.

4) In “Discussion”, many assertions are questionable and sometimes contradictory. The findings are not sufficient to explain the role of CYP46A1 deletion in modulating different behavior and learning phenotypes in the two KO mouse models. Moreover, the therapeutic relevance of Cyp46A1 cannot be supported by the data.

5)  Bibliography: several quotations are not correct.

Comments on the Quality of English Language

The manuscript is well written but minor editing of English language are required 

Author Response

We thank the reviewer for the constructive criticism. Here we attempt to address the shortcomings, with the editorial turnaround time we were given. We appreciate that this work opens further avenues of exploration, and we look forward to pursuing some of these in subsequent work.

  1. We have edited the Introduction to give more precise context behind the models used. Overall context and rationale are given in the first paragraph of the Introduction. Rationale for the 2 models is given in the 2nd paragraph of the Introduction. We have deleted the redundant reference to ‘whole body’ in favor of the term ‘global’ knockouts. Because 24S-HC is a secreted molecule, it is difficult to know whether other cells have been affected in a non-cell autonomous manner (p. 15-16). The vGLUT1-cre mice have been previously validated, and we now clarify their source (p. 3). Most studies were performed with PND100 mice, with select studies in PND400 mice. Specifically, we have previously performed 24S-HC measurements in PND100 mice, and we now cite that reference. We have also clarified age in figure legends as well as Results.
  2. The mice are better defined in the last paragraph of the Introduction, and the definitions are carried through the manuscript. We define the controls in Figure 1C in the legend. We agree that the transcriptional data are hypothesis-generating and thus preliminary; this is now acknowledged (p. 16).
  3. Statistical analysis has been added to Figure 1C. For the comparison between KO and cKO in Figure 9, we do not have sufficient statistical power for this direct comparison at PND100, and the direct comparison between panels B and C introduces age as a variable. We hope that the replication of KO results at PND400 in Figure 9C (as compared with 9A and with RNA-seq results) gives the convergent evidence needed for confidence in the robustness of the global KO result and difference from the cKO data.
  4. We agree that some of the results are paradoxical and try to point these out clearly. We have softened the language around therapeutic implications (p. 17).
  5. We have checked the reference list carefully and corrected a few formatting and placement errors. Also, in response to the editor’s query about self-citations, we note that we are among only a few groups researching 24S-HC signaling in the nervous system. We have scrutinized the references, and we believe that all are appropriate.

Reviewer 2 Report

Comments and Suggestions for Authors

This manuscript described the behavioral and transcriptome changes in the global cyp46a1 KO mice, in comparison to the conditional KO (cKO) mice in which cyp46a1 is selectively deleted in vGLUT1+ excitatory neurons.  Although 24S-HC levels are reduced by 2/3 in the hippocampus in the cKO mice, these mice did not show any behavioral deficits, in contrast to mice with global cyp46a1 KO. This manuscript is well-written and clearly presented, and discussion is sufficient. It adds new information to the field by using the cyp46a1 cKO mice that has not been reported previously. 

There are several comments that need to be addressed:

1.  The source of vGLUT1-cre mouse line is unclear. Besides the expression in glutamatergic neurons in cortex and hippocampus, is it expressed in other regions, such as amygdala, which is responsible for fear conditioning? Fig 1C shows 24S-HC level in the hippocampus, how the levels in the cortex and amygdala are affected in the cKO mice? Is cyp46a1 expressed in amygdala neurons.

2.  Regarding the age effects on behavior, are there age-dependent differences in the expression of cyp46a1/24S-HC over the period of PND45-PND400, the range of mice that were used in this study?

3.  Different 24S-HC dose (Fig. 1C) may explain the behavioral discrepancy between the global and cKO line. From Fig.9, it looks like that global cyp46a1 KO results in inhibition of de novo cholesterol biosynthesis, whereas cKO does not. Is this part of the reasons too? Is it possible that cyp46a1 has other signaling function in addition to producing 24S-HC in different neuronal populations?

Author Response

We thank the reviewer for the positive comments and for the overall assessment that the contribution will be useful and adds new information.  We respond to the critiques below.

  1. The source of the vGLUT1-cre line is now given (p. 3). B. Our original expectation was that hippocampal dependent learning would be affected because of previously published results. This expectation guided our focus on 24S levels in hippocampus. Nevertheless, CYP46a1 amygdala RNA and protein expression levels are comparable to those in hippocampus (PMID 10377398, 33192294, 35328827). vGLUT1 dependent deletion should have reduced CYP46A1 and 24SHC in amygdala (37884748). We now comment on amygdala levels and acknowledge that direct measurements from amygdala in the cKO animals await future experiments (p. 15-16).
  2. We don’t have direct data on this; however, previous work has shown that brain 24S-HC levels increase with maturation, plateauing near PND180 but with near-final levels achieved at PND30 (10377398) (p. 15)
  3. We agree that dose effects could explain differences between the full knockout and conditional knockout. This is now discussed (p. 15). Previous work has suggested that global Cyp46a1 deletion does not alter cholesterol levels despite transcriptional changes (p. 15). We agree that additional signaling functions for the enzyme are a possibility and now comment on this (p. 15).

Reviewer 3 Report

Comments and Suggestions for Authors

Behavioural effects were addressed,  including hippocampal-dependent learning as well as transcription changes evoked by 24S-HC reduction by using complete genetic deletion of cholesterol 24-hydroxylase - KO mice,  and under conditions of selective enzyme knockout in excitatory VGLUT1 cells. It should be noted that the latter model is a novel original approach introduced by the authors to study the impact of enzyme loss.   Applied methods are sound and relevant. The obtained results are presented in 9 Figures and 5 Tables.

The results show that in comparison with control WT mice constitutive KO mice demonstrated hypolocomotion, lower rearing frequency, and reduced exploration, while only the tendencies could be noted in conditional deletion mice.

Examination of hippocampal-dependent learning based on contextual fear conditioning (tone-shock pairing, auditory cue) and spatial learning in the Morris water maze, revealed increased freezing behaviour in KO mice and no significant changes in spatial learning, which suggests better memory retention.

Worth adding that the changes were only partially alleviated by the 24S-HC analogue. Concerning the selective knockout animals their learning behaviour did not match that of KO mice. Moreover, in hippocamps of KO mice Apoe and Abca1 genes were upregulated by Cyp46a1 loss whereas in animals with selective deletion of Cyp46a1 in excitatory neurons, these gene expressions were higher but not as much as in KO mice. However, the hippocampal level of 24S-HC in these mice was roughly 1/3 of WT controls. The transcriptional changes suggest therefore a dose-dependent relationship between CYP46A1, 24S-HC, and gene expression. 

The obtained results are thoroughly discussed about published data;  47 references were cited.

Conclusions are consistent with the evidence and presented arguments.

Author Response

We thank the reviewer for the succinct summary and positive evaluation.

Round 2

Reviewer 1 Report

Comments and Suggestions for Authors

The revision of data description, discussion and statistic analysis, as well as bibliography correction have improved the scientific soundness of the manuscript. For these reasons the present version of the manuscript is suitable for publication   

Comments on the Quality of English Language

There  are still few mistakes